# Aetiology of Primary Spontaneous Pneumothorax

**DOI:** 10.3390/jcm11030490

**Published:** 2022-01-19

**Authors:** Rob Hallifax

**Affiliations:** Respiratory Trials Unit, Oxford Centre for Respiratory Medicine, Churchill Hospital, NHS Trust, Oxford OX3 7LJ, UK; robert.hallifax@ndm.ox.ac.uk

**Keywords:** pneumothorax, spontaneous, aetiology, bleb, bullae, metalloproteinase

## Abstract

Air in the pleural cavity is termed pneumothorax. When this occurs in the absence of trauma or medical intervention, it is called spontaneous pneumothorax. Primary spontaneous pneumothorax typically occurs in young patients without known lung disease. However, the idea that these patients have “normal” lungs is outdated. This article will review evidence of inflammation and respiratory bronchiolitis on surgical specimens, discuss the identification of emphysema-like change (i.e., blebs and bullae), the concept of pleural porosity and review recent data on the overexpression of matrix metalloproteinases in the lungs of patients who have had pneumothorax.

## 1. Historical Perspective

The pleural cavity is a virtual space which only contains a small amount of fluid that allows lung sliding during respiration. If air is present in the pleural space, it is termed pneumothorax.

Cases of pneumothorax have been documented as far back as the 15th century, when an Ottoman surgeon described a case of pneumothorax complicating traumatic fracture of the ribs [1]. Later descriptions of pneumothorax appear in 19th-century medical literature by Laennec who referred to pneumothoraces complicating Mycobacterium Tuberculosis (TB) infection. He also acknowledged the occurrence of spontaneous pneumothorax, in the absence of TB, which he termed “pneumothorax simple” [2].

Spontaneous pneumothorax occurs in the absence of preceding trauma or iatrogenic injury. Therefore, the air present in the pleural space must have arisen from an abnormal communication between the air-containing alveolar spaces of the lung and the pleura. Spontaneous pneumothoraces are subdivided into primary spontaneous pneumothorax (PSP) that occurs in patient without known lung disease and apparently “normal lungs”; those arising in patients with known underlying lung disease (such as chronic obstructive pulmonary disease (COPD) or pulmonary fibrosis) are called secondary spontaneous pneumothoraces (SSP). In PSP, the exact pathogenesis of the abnormal communication between alveoli and pleura is not clear.

Although PSP was traditionally thought to occur in the absence of lung disease, there is increasing evidence that these patients do not have “normal lungs” [3,4]. This evidence comprises a number of sources including patient-related risk factors (such as body morphology and smoking), radiographic abnormalities, and histological changes on examination of lung resection specimens.

## 2. Aetiology of Pneumothorax

### 2.1. Body Mass Index (BMI)

PSP is more common in tall thin patients with a low body mass index (BMI) and those who are smokers [5,6,7]. After a first episode, the risk of recurrence in PSP patients is higher in patients who are taller, but only in males [5]. Female height and lower BMI were not associated with increased risk of further recurrence. A large study in Japan of 5604 pneumothorax cases found that patients who had an episode of pneumothorax had a significantly lower weight:height ratio (using the Kaup index) compared to healthy individuals [6]. Compared to national standard values, the PSP patients were ectomorphic (i.e., tall and thin) from childhood, but had a marked increase in their height between the ages of 11 and 14 years [6]. The expected increase in height gain per year in adolescence started 2 years earlier in pneumothorax patients than their increase in weight (in the standard group this peak occurred simultaneously). It is postulated that as pleural pressures increase from base to apex of the lung, individuals who are taller experience higher distension pressures at the lung apex [8]. The increased pressure may then predispose them to bleb and bullae formation. However, this speculative hypothesis has never been proven and other mechanisms (discussed below) are also likely to be required alongside this rapid growth.

### 2.2. Smoking

Smoking is a recognised risk factor for PSP. One early study from a Swedish population suggested that smoking increased the relative risk of first episode of pneumothorax by a factor of 22 in men and 9 in women [9]. Furthermore, a systematic review found that smoking cessation reduces the rate of recurrence by a factor of 4 (odds ratio of 0.26) [10]. There is evidence from surgical lung specimens that smoking causes inflammation within the lungs of patients with pneumothorax (see below).

### 2.3. Abnormal Underlying Lung

While PSP seems to occur in patients without overtly recognised lung disease, patients may have blebs or bullae seen on computed tomography (CT) scanning (see Figure 1). It is often thought that it is rupture of a bleb or bullae that causes the air to leak from the lung into the pleural space, thereby causing the pneumothorax. Indeed, most surgical procedures undertaken to reduce recurrence risk in patients who have had multiple episodes of PSP usually include apical wedge resection or “bullectomy” (“Thoracoscopy for Spontaneous Pneumothorax” in special edition [11]).

### 2.4. Blebs and Bullae

Blebs and bullae are sometimes known as emphysema-like change (ELC). Blebs were first described by Miller in 1947, as distinct from a bulla (or bullous emphysema) [12]. A bleb can be defined as an outpouching (or vesicle) of the visceral pleura caused by air in the interstitium, forming between the lamina elastica interna and externa of the pulmonary pleura, typically <1 cm in diameter, whereas a bulla (i.e., subpleural emphysematous bulla) is an airspace measuring >1 cm, which is sharply demarcated by a thin wall (no greater than 1 mm in thickness). In 1967, Reid further subdivided divided bulla into three types: “type I, a small amount of hyper-inflated lung tissue that is narrow (pedunculated) and contains no lung parenchyma; type II, a relatively smaller amount of hyper-inflated lung tissue that is broad (sessile) and usually contains vanishing lung; and type III, a large amount of hyper-inflated lung tissue extending to the pulmonary hilum, with ill-defined margins and vanishing parenchyma in each bulla” (see Figure 2) [13]. The view of a bleb on the visceral pleural surface at thoracoscopy is shown in Figure 3.

The true incidence of ELC in PSP patients (compared to the similar patient groups without pneumothorax) is not clear. One early study found these changes in 81% of non-smoking patients with PSP (predominately in the upper lobes) but not in healthy controls. Bilateral ELC were seen in 15% of patients with unilateral pneumothorax [9]. However, a larger series of 250 healthy individuals with no prior history of pleural disease undergoing thoracoscopic sympathectomy found a 6% incidence of apical blebs, which were more prevalent in slim individuals (BMI < 22 kg/m^2^) who smoked [15].

Interestingly, the use of the term bleb by radiologists is discouraged by Fleischner guidelines as it is thought to be of little clinical importance [16]. However, there is evidence that different types of bullae may have differing microscopic abnormalities [17].

### 2.5. Microscopic Abnormalities—Inflammation

Microscopic examination of lung parenchyma from 20 patients with PSP who had undergone surgical resection in a study from 1971 showed chronic distal airway inflammation: lymphocyte and macrophage infiltration with some fibrotic changes [18]. It could be that this chronic inflammation leads to the formation of ELCs in otherwise healthy lungs. However, this has not been proven. Another study found histopathologic evidence of respiratory bronchiolitis (RB) in 88.6% of patients undergoing surgery for PSP [19]. RB is the accumulation of pigmented macrophages in the lumen and walls of bronchioles. All the patients in this study were smokers, and smoking is a known cause of RB; so it is unclear whether RB is a causal step in the pathway of pneumothorax development, or simply a marker of the patient’s smoking status. Another surgical study of 126 patients found RB in only 50% of cases [17]. In the same patients, light microscopic assessment found thickening and elastofibrosis of the pleura. Reid type II bulla were found to have microscopic changes in keeping with that originally described by Miller with respect to blebs, i.e., formation of small air-containing space between lamina elastic interna and externa that contains no lung parenchyma (Figure 1) [13,17]. Electron microscopic examination showed further differences between Reid type I and type II bullae: type I bullae showed marked absence of mesothelial cells, with underlying collagen fibres clearly seen and pores of 10–20 μm in diameter were noted; type II bullae (and giant bullae) have relatively well preserved mesothelial cells with short microvilli [17]. Therefore, it could follow that these smaller “blebs” (consistent with Reid description of “type I bulla”), with absence of mesothelilal cells and nearby pores, are at greater risk of air leaking than the often larger bullae (described as Reid “type II bullae”) with intact mesothelial cells. These areas of disrupted mesothelial cells at the visceral pleural are replaced by an inflammatory elastofibrotic layer, which may contribute to an increased tendency to leak air. This idea is known as “pleural porosity”.

### 2.6. Pleural Porosity

Historically, it was postulated that it was the rupture of blebs or bulla which causes a leakage of air from the alveola to the pleural space thus creating a pneumothorax. However, if this were true, there would be evidence of ruptured blebs/bullae at medical thoracoscopy or video-assisted thoracoscopy surgery (VATS). In reality, visible air leaks from ELCs are highly variable, with many blebs or bullae remaining intact and, in some cases, no macroscopic lesions are seen at all [20]. This supports the idea of pleural porosity, i.e., that pneumothorax occurs in PSP when air leaks from thinned visceral pleural rather than rupture of a bleb or bullae. This idea was elegantly demonstrated in an experiment by Noppen et al. [21]. In this study, 12 patients with PSP were compared to 17 control subjects at thoracoscopy using fluorescein-enhanced autofluorescence. The control subjects were undergoing sympathectomy and did not have any lung disease, or previous pneumothorax. Fluorescein glows green under ultraviolet light. Patients inhaled nebulised fluorescein prior to their procedure. This study demonstrated that areas of the peripheral lung showed green subpleural fluorescence under ultraviolet light, suggesting that the inhaled fluorescein was coming close to the surface of lung at regions of the lung which otherwise appeared normal on plain white light thoracoscopy. The most abnormal lesions were seen in PSP patients. Two patients with ELC demonstrated air leak, but importantly the leakage was not directly at the site of the macroscopic blebs or bullae [21].

### 2.7. Abnormal Elastolysis

It is not clear why chronic peripheral airway inflammation should result in degradation of elastic fibres, and subsequent formation of a porous elastofibrotic layer. There is evidence of an imbalance between the protease-antiprotease and oxidant-antioxidant systems. Matrix metalloproteinases (MMP) are a zinc- and calcium-dependent endopeptidases which can damage the barrier between pulmonary epithelium and alveoli [22]. MMP-2 and MMP-9 are thought to be pathogenic in other lung diseases [23] including asthma and COPD [24]. Two studies of surgical resection specimens of patients undergoing surgery for PSP demonstrated overexpression of MMPs [22,25]. One study compared lung tissue from 15 PSP patients with that from 20 control patients (with early-stage lung cancer) [22]. Immunohistochemistry on slides of the lung tissue showed overexpression of MMP-2, -7 and -9 in the PSP patients compared to controls [22]. The other study of 91 pneumothorax patients (without a control group) also found high MMP-2 and MMP-9 expression. Moreover, patients with recurrent pneumothorax episodes had higher levels of MMP expression [25]. Alongside the overexpression of the potentially damaging MMPs, propensity for the lung tissue to become more porous may be due to the absence of other “protective” factors. Nuclear factor erythroid 2-related factor 2 (Nrf2) is a regulator of redox homeostasis and inflammatory response which may have a cytoprotective role in detoxification and chemoprevention [26]. The lung tissue of same patient group as above found higher Nrf2 expression (in alveolar type I pneumocytes) in patients without previous recurrence, suggesting a possible protective effect [26].

### 2.8. Familial Causes

Approximately 10% of patients with PSP will have a significant family history of pneumothorax [27,28]. However, this estimate is based upon two small studies: a survey of males from the Israeli Defence Force identified 15 families with a strong history of pneumothorax [27]; and another study of 102 patients and their family members in China [28].

There are a number of associated heritable conditions which predispose to pneumothorax formation including connective tissue disorders (Marfan syndrome [29], Ehlers–Danlos syndrome, or other mutations of the folliculin gene), disorders with a tendency for cyst or emphysema formation (Birt–Hogg–Dube (BHD) syndrome [30] and alpha-1 antitrypsin deficiency [31], respectively), and metabolic disorders (such as Homocystinuria [32]). Although rare, patients with a strong family history of spontaneous pneumothorax require further investigation as the pneumothorax episode may be a precursor to one identification of these disorders, which require lifelong surveillance for other severe associations such as renal cancer in BHD syndrome or aortic rupture in Marfan syndrome. CT scanning of the chest and pulmonary function testing are not currently routinely requested after uncomplicated treatment of a first episode of PSP. However, clinicians should be vigilant for PSP being the first manifestation of a systemic disease, and have a low-threshold CT scan and referral for genetic testing in patients with a family history of pneumothorax or features of another associated condition [33].

## 3. Conclusions

There is increasing evidence that the lungs of patients with PSP are not “normal” as traditionally described in textbooks. There appears to be an association between low BMI and smoking and risk of developing pneumothorax. Blebs are a common finding on CT scan on patients with pneumothorax, but the significance as a prognostic marker of pneumothorax recurrence is not yet proven. Analysis of surgical specimens has shown inflammation and respiratory bronchiolitis. More recently, matrix metalloproteinases have been identified as possible pathological factors. Further research is required to accurately define the true pathogenic pathway leading to the formation of pneumothorax in this patient group. However, identifying causative agents may allow more accurate stratification of patients at increased risk and identify possible future treatment targets.

## Figures and Tables

**Figure 1 jcm-11-00490-f001:**
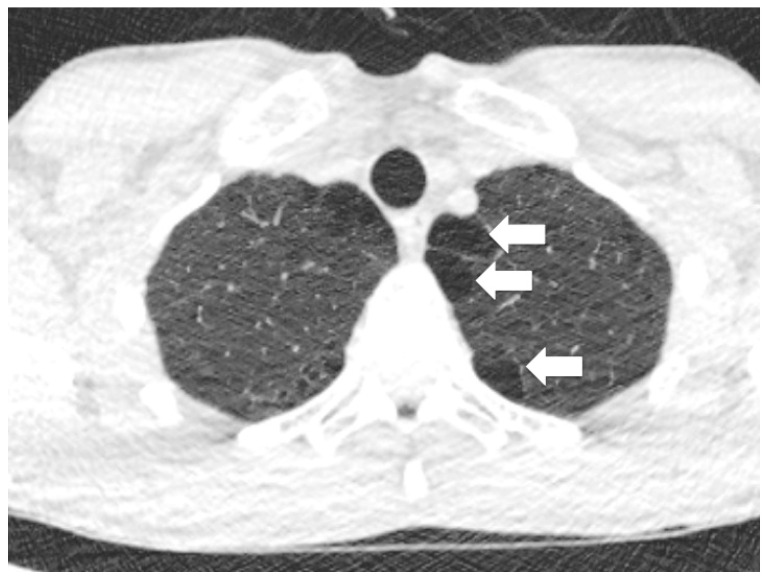
Computed tomography (CT) image demonsrates apical blebs. Arrows show multiple blebs (which are termed paraseptal emphyema when contigunous).

**Figure 2 jcm-11-00490-f002:**
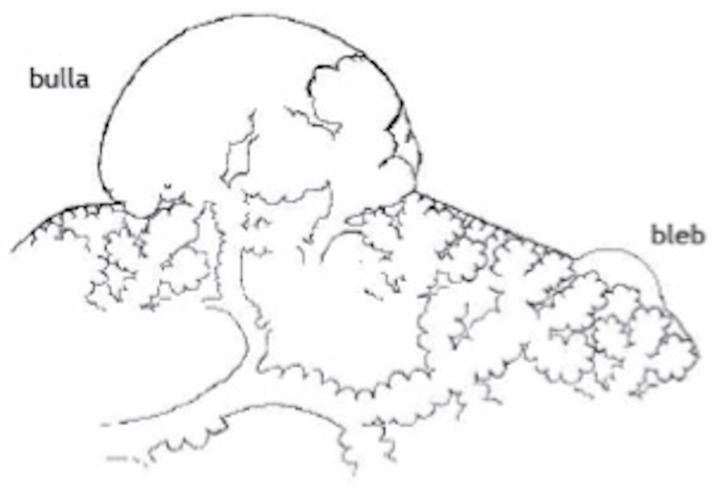
Schematic illustration of the anatomy of emphysema-like changes: a bleb and a bulla as described by Reid as type I and type II, respectively [13] (from Lyra et al. [14]).

**Figure 3 jcm-11-00490-f003:**
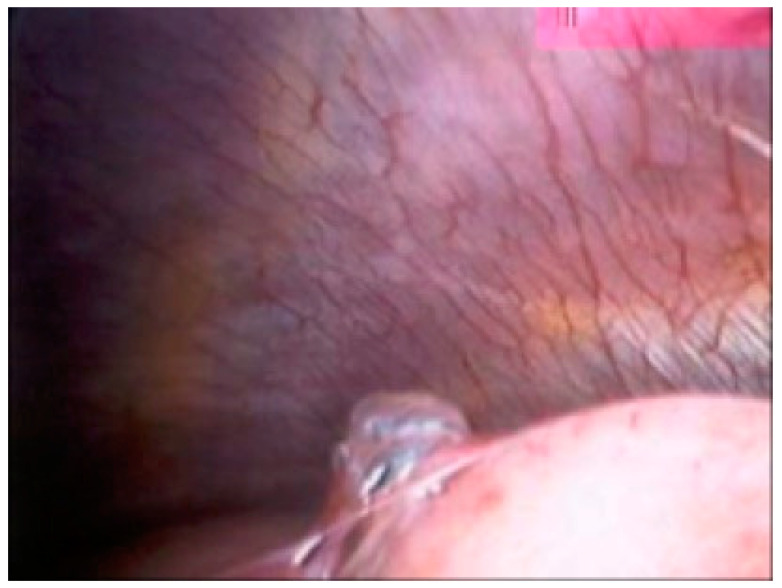
Thoracoscopic view of a bleb on the visceral pleural surface (Courtesy of M Noppen).

## Data Availability

Not applicable.

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
