# Peer review of "Aetiology of Primary Spontaneous Pneumothorax"

_jcm, 2022, doi:10.3390/jcm11030490_

Round 1

Reviewer 1 Report

This is a short but very concise manuscript that reviews the aetiology of primary spontaneous pneumothorax. The author has made a review of the most important and recent articles on the matter from different points of view: historical, epidemiological, pathophysiological, radiological, anatomopathological, molecular and genetic. I wanted to congratulate the author for the great work done. I just have a few minor comments:

  1. Section 2.1, lines 11-13: Please provide bibliographic reference where this pathophysiological theory was postulated.

  1. Section 2.3: I recommend adding a figure that illustrates the different types of bullae proposed by Reid.

  1. Section 2.3: The author states that 81% of patients with smoking history and primary spontaneous pneumothorax had emphysema-like changes in both upper lobes bilaterally. Are there data on the location and unilaterality/bilaterality of the pneumothorax in these patients?

Reviewer 2 Report

Thank you for the opportunity to review this manuscript, which briefly but quite completely describes the aetiology of primary spontaneous pneumothorax.

Strengths

- The topic is of interest and appropriate as part of the special issue

Weaknesses

- A higher attention to details could improve the quality of the article (typos, syntax mistakes)

Here are some minor comments for the author:

In paragraph 2, the author could maybe consider adding a paragraph with regard to the role of smoking history as a risk factor of pneumothorax.

Paragraph 2.5

-“This supports the alternative theory of pleural porosity, as described above (rather than ELC rupture), may be critical in pneumothorax formation.”: I am not sure I understand, could the author please check the syntax in this sentence?

-“This study demonstrated, for the first time, that areas of parenchymal abnormality (areas of subpleural fluorescence or visible fluorescence leak when inspected under ultraviolet light) at regions of the lung which otherwise appeared normal on plain white light thoracoscopy.”: could the author please check the syntax in this phrase?

Conclusions

- “There appears to be an association between low BMI and risk of developing pneumothorax - particalarly those who grow rapidly in adolescence.”: could the author please check the syntax in this sentence and correct the typo?

- “Blebs are a common finding on CT scan on patients with pneumothorax, but there signficance as a prognostic marker of pneumothroax recurrence is not yet proven.”: the author should check the spelling.

References

- Reference are indicated with different styles: the author should change the format as indicated by the Journal’s guidelines.

Round 2

Reviewer 2 Report

I reviewed the revised version of the manuscript, and I want to thank the author for the response he provided and for kindly accepting my comments.

I only have very few minor comments:

-could you please check the syntax in the first sentence in the “microscopic abnormalities” paragraph? I could not fully understand this phrase.

-there are still few typos throughout the manuscript

-is figure 3 cited in the text?